# Dietary Patterns and Nutrient Intake in University Students of Macao: A Cross-Sectional Study

**DOI:** 10.3390/nu14173642

**Published:** 2022-09-03

**Authors:** Xiaoyu Tao, Ying Shao, Donghan Xu, Yunzhi Huang, Xi Yu, Tian Zhong, Ling Wang, Sookja Kim Chung, Dong Chen, Lili Yu, Ying Xiao

**Affiliations:** 1Faculty of Medicine, Macau University of Science and Technology, Taipa, Macao SAR, China; 2Faculty of Chinese Medicine, Macau University of Science and Technology, Taipa, Macao SAR, China; 3School of Education, Zhongshan Polytechnic, Zhongshan 528404, China; 4Dr. Neher’s Biophysics Laboratory for Innovative Drug Discovery, State Key Laboratory of Quality Research in Chinese Medicine, Macau University of Science and Technology, Taipa, Macao SAR, China

**Keywords:** university students, dietary pattern, nutrient intake, cross-sectional study

## Abstract

Background: Nutritional status affects the health of the public and is one of the key factors influencing social-economic development. To date, little research on the nutritional status of the Macao university student population has been conducted. Objectives: To identify and evaluate the dietary pattern and the nutritional intake among Macao university students. Methods: The Macao students were selected by the stratified cluster random sampling method. A semi-quantitative food frequency questionnaire was used to investigate food consumption. Data were analyzed through a *t-test* and *factor analysis* by using SPSS Version 24.0. Results: A total of 1230 questionnaires were distributed. From the respondents, 1067 (86.7%) were valid. In general, we identified three major dietary patterns in this population: (1) fruit and vegetable dietary pattern, characterized by abundant consumption of fruits and vegetables; (2) grain and high fat dietary pattern, characterized as high intakes of grains and animal foods; (3) high sugar dietary pattern, characterized by a large quantity of daily sugary drinks. The average daily intake of vitamin A, thiamine, calcium, and iodine were significantly lower than the Chinese Recommended Nutrient Intake (RNI) in the subjects. **Conclusions:** The dietary pattern of Macao students is similar to that of other Asians. Surprisingly, the daily intake of vitamin A, thiamine, calcium, and iodine by Macao university students is significantly lower than the Chinese RNI.

## 1. Introduction

Generally, the age of college students is around 20-years-old, which is the period of having an optimal health status for majority of this population. With this advantage, a large number of these students give a cold shoulder to health maintenance and promotion, leading to an unhealthy lifestyle with an imbalanced dietary intake. Due to a fast-paced lifestyle with heavy course work, an increasing proportion of university students chooses to dine out with unintentional food choices. Besides, the dietary preference in making choices of food types is among the most frequent human behavior. Multiple factors at the individual, interpersonal, environmental, and societal levels influence the pattern of eating in an individual over time. Dietary patterns, in turn, influence the risk of chronic diseases and affect the quality of life [1]. Moreover, there is evidence that young adulthood is a crucial life-stage for establishing life-long healthy behaviors and habits, including healthy eating patterns [2]. A healthy eating pattern may reflect healthier food choices and nutritional dietary intake. According to the scientific reports from the Dietary Guidelines Advisory Committee, a healthy eating pattern is a diet of whole grains, vegetables, fruits, low-fat and no-fat dairy products, and lean meats, which can reduce the risk of obesity; metabolic syndrome; diabetes and its complications, such as cardiovascular disease; and some cancers [3]. The identification of dietary patterns may be helpful for understanding their relation to diet-related chronic diseases. For example, a Westernization-structure dietary pattern could increase the risk of being overweight/obesity and central adiposity. A meat-structure dietary pattern may could increase the risk of hypertension, whereas some identified Chinese and Western dietary patterns may reduce the risk of low bone mineral quality [4]. On the other hand, studies have shown that many adolescents living away from home to attend college may show numerous health-related behavioral changes, including the adopting of unhealthy dietary habits [5]. Unreasonable dietary patterns can cause greater harm to the health condition, such as being overweight and obesity. Being overweight and obesity became a pandemic and are international public health challenges. There is a global urgent need for preventing these conditions, since they are major risk factors for morbidity and premature death from a range of chronic diseases [6]. University students are a high-risk group for rapid and significant weight gain over a short period of time [7]. 

Macao is the hub of the Guangdong–Hong Kong–Macao Greater Bay Area (GBA). As the reserve force of the shaping and development of the GBA, Macao should pay great attention to monitoring of the health condition of university students in Macao. Currently, the relevant data are either incomplete or obsolete. Therefore, the present study sought to fulfill this gap. A cross-sectional study was conducted to identify and evaluate the dietary patterns and nutrient intake in university students of Macao. We used multivariate statistical techniques to identify dietary patterns through the intake of multiple interrelated food groups and captured the complexity and multidimensional nature of diet. This approach also allowed greater insight into the different patterns of food consumption that naturally occur within a population and facilitated identification of sub-groups who may be most in need of health promotion efforts [8]. The aim of this research project was to investigate the dietary patterns among university students in Macao, since there is limited research and reports on the dietary patterns among university students in Macao. Taken together, the results from the current study should be of great value for future nutritional interventions at the university-level setting to improve diet quality and, thereby, reduce the risk of lifelong chronic disease in the future generations.

## 2. Materials and Methods

### 2.1. Study Design and Participants

A cross-sectional survey was conducted through the semi-quantitative food frequency questionnaire (SQFFQ) during October 2020. The entire study lasted for 25 consecutive days. The university students of Macao were selected by the stratified cluster random sampling method. Both residential and non-residential students were included in the study. A total of 1230 questionnaires were issued, of which 1067 were retrieved, and 163 invalid questionnaires due to missing data or extreme outliers were removed. The contents of the SQFFQ consisted of three major parts, including demographic information of the subjects, the frequency and amount of food consumed in the past one month, as well as the informed consent form. Among the questionnaire, a special SQFFQ including 48 food items was developed based on the *Survey of Food Consumption Status of Chinese Residents (2017)* [9]. In addition, the food atlas corresponding to each food item (containing portion size) was compiled and handed out to the students together with the questionnaire to facilitate the assessment of their consumed amount of each food item. The body weight and height of the respondents were collected based on their report as well.

### 2.2. Data Processing

The calculation of nutrient and energy intake for each food item in the SQFFQ was based on the *China Food Composition* [10] and *Standards Tables of Food Composition in Japan 2015* [11]. The specific nutrient intake of the respondents was calculated as follows. The nutrient composition of each food listed in the SQFFQ was sorted out before data processing. In this study, food intake was calculated in the edible state, and the specific process of the nutrient composition of various foods included raw and cooked food conversion, determination of food composition values, and determination of the proportion of edible portions. Then, the average daily intake of each food item and the average daily intake of nutrients were calculated. The average daily intake of each food was calculated separately for each individual firstly. The specific nutrient contents of each food item were then calculated separately. On this basis, the sum of all food intake pairs corresponding to the nutrient intakes of each individual respondent was calculated, and the results obtained were then converted into the average daily intakes of nutrients for the respondents. The data for daily energy intake being much lower or higher than the normal range, such as 800–4500 kcal for males and 500–3500 kcal for females, were considered as outliers and excluded, respectively [12].

### 2.3. Statistical Analyses

A two-tailed (α = 0.05) statistical verification was performed. All data were analyzed via SPSS Version 24.0 (SPSS Inc., IBM) by a *t-test* and *factor analysis*. The data are presented as the *mean* and *standard deviation (SD)* for the normal-distributed and *Percentile* for the skew-distributed ones. The evaluation of the dietary patterns of the respondents was conducted by a *f**actor a**nalysis*. Orthogonal rotations using Kaiser’s criterion were utilized to establish the number of factors (we used an eigenvalue >1 criterion). The dietary patterns were named according to the magnitude of the factor scores and the interpretability of the overall diet.

### 2.4. Outcome Measures

To facilitate the calculation of the results, the density of liquid foods was approximated as the density of drinking water, and the unit of volume (mL) was converted into the mass unit as a “*gram*”. The daily nutrient intake was compared with the *China dietary reference intakes (DRIs)* in the subjects. The evaluation of daily nutrient intake was conducted according to the method adopted from the DRIs [13]. Energy and 17 different kinds of nutrients were taken into account in the study, including protein, fat, carbohydrate, cholesterol, vitamin A, thiamine, riboflavin, ascorbic acid, calcium, iron, phosphorus, magnesium, zinc, potassium, sodium, iodine, and selenium [10,14,15,16], and the respondents’ energy intake was evaluated with consideration of the level of light physical activity [17].

## 3. Results

### 3.1. General Demographic Characteristics of the Subjects

A total of 1230 questionnaires were distributed, and 1168 (95.0%) were collected, of which 1067 (86.7%) were valid and were used for data analysis. Among the 1067 valid respondents, 430 (40.3%) were males, and 637 (59.7%) were females. The average age of the respondents was 20 years.

### 3.2. Body Weight, Height and BMI of the Subjects 

The average body weight students was 70 ± 13 (kg) and 54 ± 10 (kg) for male and female students, respectively. The average height was 1.76 ± 0.06 (m) and 1.64 ± 0.05 (m) for male and female students, respectively. The average BMI was 22.5 ± 3.8 (kg/m^2^) and 20.1 ± 3.6 (kg/m^2^) for male and female students, respectively. According to the *Guidelines for the Prevention and Control of Overweightness and Obesity in Chinese Adults* [18], the respondents were grouped into underweight, normal weight, overweight, and obese. The results are shown in Table 1.

### 3.3. The Average Daily Consumption of Different Food Categories

The average daily consumption of different food categories among respondents are shown in Table 2.

### 3.4. Dietary Patterns

Beverages and others (syrup drinks) were integrated into one category, according to their common characteristics of food, as sugary drinks; drinking water was excluded in the process of dietary patterns’ extraction, since it has less contribution to daily nutrient intake; a total of 9 groups of food were included in the factor analysis.

The Kaiser–Meyer–Olkin (KMO) measures indicated the correlation between the food groups (KMO = 0.687). The Bartlett test of Sphericity for food groups among respondents was highly significant (p < 0.001), indicating a homogeneity of variance.

According to the Kaiser Criterion, three factors for respondents were extracted by factor analysis for different food groups. The results are shown in Table 3. The three dietary patterns were named the fruit and vegetable dietary pattern, grain and high fat pattern, and high sugar pattern. The fruit and vegetable dietary pattern was characterized by a high consumption of fruits and vegetables, such as cabbage, broccoli, apples, and pears; the grain and high fat dietary pattern was characterized by grain and animal-sourced foods, such as rice, bread, corn, fish, pork and beef; the high sugar dietary pattern was characterized by sugary drinks, such as juice, soft drinks, and milk tea. The major dietary pattern of the subjects was close to the oriental dietary pattern [18]. Besides, the distributions of three dietary patterns between males and females were significantly different (*p* < 0.05). The results are shown in Table 4.

### 3.5. Assessment of Nutrient Intake

The proportion of energy provided by protein was slightly higher that the recommended proportion. In addition, all the respondents tended to have a slightly inadequate intake of some micronutrients, such as vitamin A, thiamine, calcium, sodium, and iodine. Furthermore, the average daily intake of all nutrients was within the tolerable upper intakes (UL). The results are shown in Table 5. Besides, 10 major nutrients [16] were further evaluated by the cut-off point method. The results are shown in Table 6.

## 4. Discussion

Only about 60% of the subjects ranged within the normal BMI, and the rest were either overweight/obese or underweight. Almost one-third of females were underweight, but close to a quarter of males were overweight/obese, which implied that more female students might tend to lose body weight for a “good outlook” of their shape, while male students did not pay enough attention to their weight control. Without a relevant survey, at this moment, we are unlikely to reveal the reasons for the higher proportion of their abnormal BMI.

The results showed that for all respondents, the intake of animal foods was slightly higher, while the intake of soy and nuts, milk and dairy products, and vegetables and fruits were all lower. The findings of this study are common among university students due to the level of socio-economic development, consumer values, and the psychological characteristics of university students. The results of the factor analysis showed that male respondents were more likely to choose cereals and animal food groups in their diets; female respondents were more likely to choose fruits and vegetables in their diets; after consolidating the data and combining the average daily food consumption data and the factor loadings of the respondents, it can be seen that the respondents’ choice of food groups and the amount of food consumed were mainly cereals and grains, supplemented by animal food groups, vegetables, and fruits. In this study, the continuation of such food consumption habits by the respondents posed a risk of developing several non-communicable chronic diseases.

Both the KMO test and Bartlett’s sphericity test demonstrated the suitability of the data for factor analysis. According to the Kaiser Criterion, three factors were identified in the case of setting the eigenvalue over 1. Furthermore, three typical dietary patterns in accordance with oriental dietary habits were identified by the loadings and interpretability of factors. The identified dietary patterns were named according to their characteristics. However, a factor analysis is susceptive to being subjective.

The respondent’s protein intake was slightly above the appropriate range, which may be due to the participants’ high consumption of animal-source foods. The fat intake of the respondents was in a suitable range, which is likely to be an underestimated result, because the consumption data of the cooking oil were not included in the SQFFQ. At the same time, the findings of vitamin and mineral intake of respondents were consistent with other domestic studies in China [19]. The evaluation results of the vitamin intake of respondents showed that the average daily intake of vitamin A and thiamine in the participants were seriously insufficient, and more than 90% of the respondents could not reach the EAR. Vitamin A is an essential fat-soluble vitamin needed to function in various body systems, and it plays important roles in vision, growth, and immune function [20]. Vitamin A deficiency results from a dietary intake of vitamin A that is inadequate to satisfy physiological needs. It may be exacerbated by high rates of infection, especially diarrhea and measles. It is common in developing countries, but it is rarely seen in developed countries [21]. Besides, thiamine, also known as Vitamin B1, helps the body function by taking energy from food and turning it into fuel for the brain, nerves, and heart. A thiamine deficiency can result in several health problems, including confusion, seizures, shortness of breath, brain disease, coma, and more [22]. It is worth noting that thiamine deficiency is still an important health issue in developing countries, especially in Asia [23]. The respondents’ deficiencies in these two nutrients also coincided with their consuming less fruits and vegetables and more refined grains. As for the mineral intake, the results revealed that the average daily intake of calcium and sodium of the respondents was slightly lower than the RNI, and more than 80% did not reach the EAR. Calcium deficiency is a worldwide nutritional deficiency public-health problem, and similar to other countries, calcium deficiency is common in China [24]. Calcium deficiency can cause osteoporosis, hypertension, colon cancer, and kidney stones [25]. Thus, for the respondents, it is equally important to pay attention to calcium supplementation in the diet, especially to increase their intake of dairy products or look for a calcium supplement that also contains vitamin D, as these two nutrients work together and increase the amount of calcium that the body can utilize. Of course, sun exposure is by far the best way to boost vitamin D levels, particularly because very few foods contain significant amounts [26]. However, the average daily intake of sodium among the respondents in the results could have been underestimated, since the main sources of sodium were condiments such as salt, soy sauce, and oyster sauce, which were not included in the SQFFQ. Besides, nearly 80% of the respondents did not reach the EAR for the daily intake of iodine, and the average daily intake of iodine was significantly insufficient. It is worth noting that in Mainland China, thanks to the full implementation of the mandatory salt iodization policy and the special district water improvement projects, the iodine deficiency disease epidemic is rarely seen nowadays. However, the China Center for Disease Control and Prevention performed the population study in four coastal provinces in China. According to their results, the iodine intake from the diet cannot meet the daily needs without iodized salt, even for these coastal area residents with a high consumption on iodine-rich seafoods [27]. Since the compulsory iodine salt policy is not fully implemented in Macao SAR, there are both iodized or non-iodized salts on the market, and the cost of non-iodized salt is lower, the low daily intake of iodine may indeed exist to some extent. In Hong Kong, which has a very similar living environment to Macao, similar with Macao, the Hong Kong Government has not implemented a mandatory salt iodization policy either. Furthermore, the Center for Food Safety of Hong Kong found that the median daily intake of iodine for adults in Hong Kong was 44 μg in 2011, and 93% of Hong Kong citizens had lower dietary iodine intake than the recommended amount (150 μg/day) [28]. Besides, the Department of Health of Hong Kong released the Iodine Survey Report in 2021, and face-to-face questionnaire interviews and spot urine iodine tests were conducted for the participants, while thyroid ultrasonography was also performed for school-aged children. Based on the survey results, the iodine status of school-aged children and pregnant women taking iodine supplements at an average daily intake of equal to or above 150 mg per day are both classified as adequate [29]. For future studies, the urinary iodine concentrations can be used to evaluate the iodine status among Macao residents, so as to further verify the above conclusions. In summary, the university students in Macao have a significantly insufficient daily intake of vitamin A, thiamine, calcium, and iodine. Dietary interventions should be considered to prevent nutritional deficiencies and to ensure a balanced diet among university students in Macao.

## 5. Conclusion

In conclusion, the dietary pattern of Macao students is similar to that of other Asians. Surprisingly, the daily intake of vitamin A, thiamine, calcium, and iodine by Macao university students is significantly lower than the Chinese RNI. Besides, the present study had certain limitations. Firstly, we calculated the BMI using self-reported values, which are frequently used in large-scale cohort studies, and we are aware that the self-reported values are less reliable. These results are merely descriptive, since BMI was not the primary outcome parameter. Secondly, although we used a validated SQFFQ, measurement error is a recognized limitation of SQFFQ, because of its reliance on the respondent’s memory for collecting information for as far back as 3 months. 

## Figures and Tables

**Table 1 nutrients-14-03642-t001:** Distribution of respondents’ BMI classification.

Category	Male	Female
	*n* * (%)	*n* * (%)
Underweight	39 (9.1)	210 (33.0)
Normal weight	276 (64.2)	377 (59.2)
Overweight	81 (18.8)	28 (4.4)
Obese	34 (7.9)	22 (3.4)
Total	430	637

* *n*: number of people.

**Table 2 nutrients-14-03642-t002:** The average daily consumption of various food groups in the subjects (g).

Food Category	Mean ± SD ^a^	P5 ^b^	P25 ^c^	P50 ^d^	P75 ^e^	P95 ^f^
Grains	274 ± 124	96	183	276	333	485
Soy and nuts	20 ± 27	0	3	11	25	67
Vegetables	177 ± 165	15	71	131	229	486
Fruits	138 ± 149	0	38	96	185	417
Animal foods	236 ± 149	66	132	201	297	527
Milk and dairy products	252 ± 232	66	132	201	297	527
Fungus and Algae	27 ± 37	0	1	14	38	91
Beverages	195 ± 236	0	42	127	264	597
Drinking water	820 ± 667	0	250	750	1238	2000
Others (syrup drinks)	19 ± 37	0	0	7	29	86

^a^ SD: standard deviation; ^b^ P5: 5th percentile; ^c^ P25: 25th percentile; ^d^ P50: 50th percentile; ^e^ P75: 75th percentile; ^f^ P95: 95th percentile.

**Table 3 nutrients-14-03642-t003:** Factor loadings of the food category for the three dietary patterns.

Food Category		Dietary Pattern	
	Fruit & Vegetable	Grain & High Fat	High Sugar
Fruits	0.698	0.029	−0.008
Vegetables	0.677	0.366	−0.083
Nuts	0.443	−0.223	0.237
Fungus and Algae	0.426	0.083	0.407
Soybeans and soy products	0.404	0.302	0.298
Grains	−0.015	0.789	−0.112
Animal food	0.200	0.653	0.302
Sugary drinks	−0.287	0.260	0.716
Milk and dairy products	0.246	−0.147	0.596

The rotation method, orthogonal rotation method with Kaiser normalization and rotation, converged in 14 iterations, which were used for obtaining the factor loadings.

**Table 4 nutrients-14-03642-t004:** Gender distribution of three dietary patterns.

Gender	Dietary Pattern			Total
	Fruit & Vegetable	Grain & High Fat	High Sugar	
	*n* * (%)	*n* * (%)	*n* * (%)	
Male	91 (21.2)	219 (50.9)	120 (27.9)	430
Female	255 (40.0)	166 (26.1)	216 (33.9)	637

* *n*: number of people.

**Table 5 nutrients-14-03642-t005:** The average daily nutrient intake compared with the levels of DRIs (DRIs: Dietary Reference Intakes).

Energy or Nutrients	Gender	Mean	EAR ^a^	RNI ^b^	AMDR ^c^	PI ^d^	UL ^e^
Energy (kcal)	Male	2170		2250			
	Female	1705		1800			
Protein (g)	Male	93	60	65			
	Female	71	50	55			
Fat (g)	Male	24			20–30		
	Female	24			20–30		
Carbohydrate (g)	Male	59			50–65		
	Female	59			50–65		
Vitamin A (μgRAE ^f^)	Male	307	560	800			3000
	Female	274	480	700			3000
Thiamine (mg)	Male	0.7	1.2	1.4			
	Female	0.6	1.0	1.2			
Riboflavin (mg)	Male	1.2	1.2	1.4			
	Female	1.1	1.0	1.2			
Ascorbic acid (mg)	Male	86	85	100		200	2000
	Female	85	85	100		200	2000
Calcium (mg)	Male	478	650	800			2000
	Female	454	650	800			2000
Iron (mg)	Male	23	9	12			40
	Female	18	15	20			40
Phosphorus (mg)	Male	1464	600	720			3500
	Female	1181	600	720			3500
Magnesium (mg)	Male	356	280	330			
	Female	307	280	330			
Zinc (mg)	Male	22	10.5	12.5			40
	Female	18	6.0	7.5			40
Potassium (mg)	Male	2279		2000		3600	
	Female	2027		2000		3600	
Sodium (mg)	Male	1308		1500		2000	
	Female	1257		1500		2000	
Iodine (μg)	Male	70	85	120			600
	Female	59	85	120			600
Selenium (μg)	Male	60		60			400
	Female	51		60			400

^a^ EAR: Estimated average requirement. A nutrient intake value that is estimated to meet the requirement of half of the healthy individuals in a group. ^b^ RNI: Recommended nutrient intakes. This value is a goal for individuals and is based upon the EAR. It is the daily dietary intake level that is sufficient to meet the nutrient requirement of 97~98% of all healthy individuals in a group. ^c^ AMDR: Acceptable Macronutrient Distribution Range. This value is the range of intake for a particular energy source that is associated with reduced risk of chronic disease, while providing intakes of essential nutrients. ^d^ PI: Proposed Intakes for Preventing Non-communicable Chronic Disease. ^e^ UL: Tolerable upper intakes. The highest level of daily nutrient intake that is likely to pose no risks of adverse health effects to almost all individuals in the general population. ^f^ RAE: Retinol activity equivalents. The calculation of RAE is each μg RAE corresponds to 1 μg retinol, 2 μg of β-carotene in oil, 12 μg of “dietary” beta-carotene, or 24 μg of the three other dietary provitamin-A carotenoids.

**Table 6 nutrients-14-03642-t006:** Distribution of major nutrient intakes in the subjects.

Nutrients	Index	Male		Female		All %
		n	%	n	%	
Vitamin A	<EAR	393	91.4	606	95.1	93.6
	EAR~RNI	25	5.8	24	3.8	4.6
	≥RNI	12	2.8	7	1.1	1.8
Thiamine	<EAR	398	92.6	592	92.9	92.8
	EAR~RNI	20	4.7	25	3.9	4.2
	≥RNI	12	2.8	20	3.1	3.0
Riboflavin	<EAR	226	52.6	319	50.1	51.1
	EAR~RNI	55	12.8	106	16.6	15.1
	≥RNI	149	34.7	212	33.3	33.8
Ascorbic acid	<EAR	268	62.3	391	61.4	61.8
	EAR~RNI	32	7.4	39	6.1	6.6
	≥RNI	130	30.2	207	32.5	31.6
Calcium	<EAR	347	80.7	540	84.8	83.1
	EAR~RNI	39	9.1	50	7.8	8.3
	≥RNI	44	10.2	47	7.4	8.6
Iron	<EAR	7	1.6	235	36.9	22.7
	EAR~RNI	22	5.1	193	30.3	20.1
	≥RNI	401	93.3	209	32.8	57.2
Phosphorus	<EAR	8	1.9	39	6.1	4.4
	EAR~RNI	12	2.8	43	6.8	5.2
	≥RNI	410	95.3	555	87.1	90.4
Magnesium	<EAR	149	34.7	313	49.1	43.3
	EAR~RNI	65	15.1	107	16.8	16.1
	≥RNI	216	50.2	217	34.1	40.6
Zinc	<EAR	34	7.9	16	2.5	4.7
	EAR~RNI	31	7.2	19	3.0	4.7
	≥RNI	365	84.9	602	94.5	90.6
Iodine	<EAR	321	74.7	519	81.5	78.7
	EAR~RNI	38	8.8	51	8.0	8.3
	≥RNI	71	16.5	67	10.5	12.0

## Data Availability

The data presented in this study are available on request from the corresponding author. The data are not publicly available due to confidentiality concerns.

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
