# Peer review of "Dietary Patterns and Nutrient Intake in University Students of Macao: A Cross-Sectional Study"

_nutrients, 2022, doi:10.3390/nu14173642_

Round 1
Reviewer 1 Report
Nutrients_ Comments to the Author_10.3390_August 2022
Comments for the Author:
1. From your 1067 candidates for this survey, how many are residential students and how many are non residential students? Or it is a mixture of both?
2. If the data were from a mixture of students, do you anticipate the outcome would be different as the residential students would have exposed to a balanced diets for the meals provided by the University catering?
3. Line 38 – what is GBA referring to?
4. 2.2 Data Processing (Line 68) – bibliography citation in foreign language is best suit with some examples of how to calculate nutrients in Table 5, say Ascorbic Acid in male (86 mg) and female (85 mg)? Line 83 ‘.. China dietary reference intake (DRI)’ Does it mean that all measurements were referenced to Asia Populations including BMI calculations? The website provided is apparently inaccessible in English format.
5. Line 129 – Milk tea – sugars added to tea by the manufacturers? Are there any sold without added sugar and any further breakdown in analysis?
6. Table 6 – it is not clear how you can estimate the number of male and female subjects that had, for example, Vitamin A <EAR, EAR-RNI and ≥RNI? Why the number of observations varies in each of the nutrient and they were sampled from the same questionnaire? The question is similar to question 5. It requires a caption in the Method Section to guide the audience to understand the description of the tables.
7. Line 174- processed food like restaurant food and fast food outlets are high in salt content and largely use non-iodised for cooking.
8. Line 182 – HKSAR has a regular survey on community iodine nutritional status. In recent report 2021, it was stated that the community is iodine adequate. On the contrary, Macao has not had any published official data to confirm his iodine nutritional status of his citizen.
9. Line 184, the price differential between iodised and non-iodised salt must be high as a deterrent of forcing citizen to buy cheaper non-iodised salt even evidence have been shown the benefits of using iodised salt for cooking is warranted.
10. Line 185 – please ensure information is current and accurate. Department of Health, HKSAR has recently published an iodine survey report in 2021 citing iodine nutrition in his community is ‘adequate’ this is in contrast to your citation that intake of iodine in his adult environment is inadequate. Please update information and amend discussion to reflect the accuracy.
11. The discussion is largely related to Iodine status while there is no mention of the importance of other nutrients in the study which is rather unfortunate. Are these students on any vitamin supplements and how do they affect students overall performance?
Author Response
Please see the attachment, thank you.

Reviewer 2 Report
Reviewer’s results to the article of nutrients-1868512-peer-review-v1
Major comments
This is the article to show the results of studying current situation of nutrients intakes among the university students of Macao. The authors used asking methods of totally 1,230 questionnaires. Their results showed that the responders had deficiencies of vitamin A, thiamine, calcium, and iodine compared with Chinese Recommended Nutrient Intake (CRNI). Among their results, the authors classified three categories of eating pattern of food intake according to the Kaiser Criterion:1, Fruit & vegetable, 2, grain & high fat, and 3, high sugar. However, the authors did not compare these food eating patterns with nutrients amounts. In the other words, they could not interpret the food eating pattern to nutrients intake. This might be critical problems because the reason why the authors utilized eating pattern to analyze without nutrients aspects. In addition, the readers would know the following two: 1, eating behavioral pattern and analyses because what kinds of foods are related with nutrients deficiencies, 2, what characteristics do students who have deficiency of vitamin A, thiamine, calcium, and iodine because the readers must know and would identify who are at risk of these deficiencies. From these aspects, three eating patterns and characteristics must be related with nutrients intake patterns.
In conclusion, the authors must connect the results of nutrients with eating patterns and characteristics they have, then this article might be considered to review.
Minor comments
1. It is not described when the study was conducted: before 2,000 or after that?
2. The demographics of university students are not described such as they are living alone or with family, how much of their family income to know the subjects are average in Macao or not.
Author Response
Please see the attachment, thank you.

Round 2
Reviewer 1 Report
please refer to the information provided below for updating Discussion Section on Iodine nutrition in Hong Kong :
https://www.news.gov.hk/eng/2021/08/20210831/20210831_170359_758.html
Author Response
Thank you for your advice and we have added references to lines 263-268 in the manuscript.
Reviewer 2 Report
Reviewer’s results to the article of nutrients-1868512-peer-review-v2
Major comments
I would respond my comments (green letters) the authors replying or corrections according to my comments for version-1 (blue letters) as the follows:
1. Among their results, the authors classified three categories of eating pattern of food intake according to the Kaiser Criterion:1, Fruit & vegetable, 2, grain & high fat, and 3, high sugar. However, the authors did not compare these food eating patterns with nutrients amounts. In the other words, they could not interpret the food eating pattern to nutrients intake. This might be critical problems because the reason why the authors utilized eating pattern to analyze without nutrients aspects.
⇒ The authors added in lines between 220 – 222 that “The respondents' deficiencies in these two nutrients also coincided with their dietary patterns, as they consumed less fruit & vegetable 221 and more refined grains”. However, they did not show any evidence in their result. So they could not prove association between their eating pattern (grains eating pattern) and deficiency of vitamin A and thiamin. So they did not respond to my question why the responders of grain eating pattern had these vitamin deficiencies.
2. In addition, the readers would know the following two: ・・・2, what characteristics do students who have deficiency of vitamin A, thiamine, calcium, and iodine because the readers must know and would identify who are at risk of these deficiencies.
⇒ The authors also did not answer to my question what characteristics do students who have deficiency of vitamin A, thiamine.
In conclusion, the authors must connect the results of nutrients with eating patterns and characteristics they have, then this article might be considered to review.
Minor comments
1. It is not described when the study was conducted: before 2,000 or after that?
⇒The authors added the study period in line 75 that “during October 2020.”
However, my concern is what days of October must be described in the scientific study.
2. The demographics of university students are not described such as they are living alone or with family, how much of their family income to know the subjects are average in Macao or not.
⇒The authors did not add also in the version-2.
